# Reduced Glycolysis and Cytotoxicity in *Staphylococcus aureus* Isolates from Chronic Rhinosinusitis as Strategies for Host Adaptation

**DOI:** 10.3390/ijms25042229

**Published:** 2024-02-13

**Authors:** Lorena Tuchscherr, Sindy Wendler, Rakesh Santhanam, Juliane Priese, Annett Reissig, Elke Müller, Rida Ali, Sylvia Müller, Bettina Löffler, Stefan Monecke, Ralf Ehricht, Orlando Guntinas-Lichius

**Affiliations:** 1Institute of Medical Microbiology, Jena University Hospital, 07747 Jena, Germany; sindy.wendler@med.uni-jena.de (S.W.); rida.ali@med.uni-jena.de (R.A.); bettina.loeffler@med.uni-jena.de (B.L.); 2Systems Biology and Bioinformatics Unit, Leibniz Institute for Natural Product Research and Infection Biology-Hans Knöll Institute, 07745 Jena, Germany; rakesh.santhanam@yahoo.com; 3Department of Otorhinolaryngology, Jena University Hospital, 07747 Jena, Germany; juliane.priese@med.uni-jena.de (J.P.); orlando.guntinas@med.uni-jena.de (O.G.-L.); 4Leibniz Institute of Photonic Technology (IPHT), 07745 Jena, Germany; annett.reissig@leibniz-ipht.de (A.R.); elke.mueller@leibniz-ipht.de (E.M.); monecke@rocketmail.com (S.M.); ralf.ehricht@uni-jena.de (R.E.); 5InfectoGnostics Research Campus, 07743 Jena, Germany; 6Institute of Immunology, University Hospital Jena, 07743 Jena, Germany; sylvia.mueller@med.uni-jena.de; 7Institute of Physical Chemistry, Friedrich-Schiller University, 07743 Jena, Germany

**Keywords:** chronic rhinosinusitis, microbiome, *Staphylococcus aureus*

## Abstract

Chronic rhinosinusitis (CRS) is a multifactorial infection of the nasal cavity and sinuses. In this study, nasal swabs from control donors (N = 128) and patients with CRS (N = 246) were analysed. Culture methods and metagenomics revealed no obvious differences in the composition of the bacterial communities between the two groups. However, at the functional level, several metabolic pathways were significantly enriched in the CRS group compared to the control group. Pathways such as carbohydrate transport metabolism, ATP synthesis, cofactors and vitamins, photosynthesis and transcription were highly enriched in CRS. In contrast, pathways related to lipid metabolism were more representative in the control microbiome. As *S. aureus* is one of the main species found in the nasal cavity, staphylococcal isolates from control and CRS samples were analysed by microarray and functional assays. Although no significant genetic differences were detected by microarray, *S. aureus* from CRS induced less cytotoxicity to lung cells and lower rates of glycolysis in host cells than control isolates. These results suggest the differential modulation of staphylococcal virulence by the environment created by other microorganisms and their interactions with host cells in control and CRS samples. These changes were reflected in the differential expression of cytokines and in the expression of Agr, the most important quorum-sensing regulator of virulence in *S. aureus*. In addition, the CRS isolates remained stable in their cytotoxicity, whereas the cytotoxic activity of *S. aureus* isolated from control subjects decreased over time during in vitro passage. These results suggest that host factors influence the virulence of *S. aureus* and promote its adaptation to the nasal environment during CRS.

## 1. Introduction

Chronic rhinosinusitis (CRS) is a long-lasting inflammation of the respiratory tract that persists for more than 12 weeks [1,2,3]. Clinically, CRS can be divided into CRS without nasal polyposis (CRSsNP) and CRS with nasal polyposis (CRSwNP). Symptoms of CRS include nasal congestion, nasal discharge, facial pain and a decreased sense of smell [4]. Despite research, the cause of CRS remains uncertain. It affects up to 10% of the world’s population during their lifetime.

Numerous risk factors contribute to the development of CRS, including obstruction, mucociliary clearance problems, osteitis, microbes, biofilms, superantigens, immune dysfunction, genetic factors and inhaled irritants [4,5,6]. The EPOS guidelines rely on symptoms and endoscopic/radiographic findings for the primary evaluation and management of CRS. Recent studies highlight the regional diversity of CRS patients from Europe, Asia and Oceania based on cytokine profiles, eosinophilic/neutrophilic patterns and *Staphylococcus aureus* enterotoxin-specific IgE expression [5]. Eosinophilia and obstruction of the lower respiratory tract are associated with the impaired production of IL-10 in response to staphylococcal enterotoxin B and alpha toxin [7,8].

The relationship between CRS and the microbiome composition has been extensively studied. Research is investigating this link by examining the microbiome composition in both controls and individuals with CRS, focusing on host–pathogen interactions [9,10]. However, nasal microbiome analysis is complex and controversial. Discrepancies often arise from different sampling techniques, sample types and locations. Notably, one study identified *Corynebacterium* sp., *Staphylococcus aureus*, *Moraxella* sp., *Streptococcus* sp. and *Haemophilus* sp. as prominent genera in CRS patients and control donors across countries [11]. In addition, several studies have found differences in the abundance of bacterial genera between controls and CRS patients [12]. The composition, stability and resilience of the sinonasal microbiota depend on demographics, geography, clinical aspects and previous therapies [10].

*S. aureus* colonizes anterior nares in 20% to 80% of healthy individuals [13] and is a relevant prevalent species in the nostrils of 25% to 50% of patients with CRS [14,15]. This colonization can lead to invasive infections in other tissues [16]. Serum studies of CRS patients have shown high levels of antibodies to toxic shock syndrome toxin 1 (TSST-1) and Panton–Valentine leukocidin (PVL). However, genes for staphylococcal antigens showed no correlation with corresponding serum anti-staphylococcal IgG [17]

As *S. aureus* is a significant colonizer in the nasal areas and a common CRS pathogen, it is unclear which nasal factors lead to its shift from colonizer to pathogen [6,18,19,20].

The interaction between *S. aureus* and epithelial cells is a critical aspect of the host–pathogen interaction and plays a significant role in the initiation and progression of infections. One important aspect of this interaction is the role of glycolysis in the host defence against *S. aureus*. Glycolysis is a metabolic pathway that converts glucose into energy for various immune functions, such as phagocytosis, cytokine production and pathogen clearance [21,22,23].

In our study, 374 samples (128 controls, 246 CRS patients) were examined using both culture and culture-independent methods, in addition to metagenomics, to identify potential pathways involved in CRS. While the *S. aureus* genetic profiles were largely comparable between control and CRS donors, CRS strains showed a significant reduction in their ability to induce cytotoxicity and glycolysis in lung cells. Interestingly, subcultivation experiments showed a decrease in virulence in control *S. aureus* isolates, a phenomenon not observed in isolates from CRS patients. While our study was a local study, the insights gained provide a valuable understanding of the role of microbes, specifically *S. aureus*, in the intricate pathogenesis of CRS.

## 2. Results

### 2.1. Control Subjects’ and CRS Patients’ Characteristics

The characteristics of the control subjects and patients with CRS are described in Table 1. There were slightly more male participants (62%) than female participants (48%). The gender distribution was similar in the control and CRS groups. Smokers were more frequent in the control group (Pearson’s chi-square test *p* = 0.029). Forty-three percent of participants had allergies, but there was no difference between the groups. Non-steroidal anti-inflammatory drug (NSAID)-exacerbated respiratory disease was seen only in the CRS group (7.3%) (Pearson’s chi-square test *p* = 0.002). Asthma was more common in CRS patients (18.7%) than in controls (5.4%) (Pearson’s chi-square test *p* = 0.003), while diabetes was more common in controls (12.5%) than in patients (3.7%). One third of the CRS patients had nasal polyps (CRSwNP; 32.5%) and two thirds had no polyps (CRSsNP; 67.5%). In addition, the characteristics are presented separately for women (Appendix A) and men (Appendix A). The analysis by gender showed that the different frequency of smokers between both groups was related to the male study participants. There were more male smokers in the control group (36.8%) than in the CRS group (14.4%; Pearson’s chi-square test *p* = 0.016). Otherwise, there were no gender differences between the female and male participants with regard to the characteristics of the patients.

### 2.2. CRS and Control Donors Had Similar Nasal Microbiomes

To characterise the microbiomes of all samples, 128 swabs from control donors and 246 swabs from patients with CRS were collected. Conventional culture and shotgun metagenomic sequencing were used to identify microbial species. No systematic differences in cultured species were found between samples isolated from control and CRS donors (Figure 1A; Appendix A, Fisher test). Only the presence of *Neisseria* sp. was significantly associated with the control group (Figure 1A, Appendix A). *Neisseria* sp.’s antibiotic susceptibility might explain its low presence in CRS due to prior treatment [11,24]. We examined CRS samples with and without antibiotics. Few CRS patients (4.9%) received antibiotics (Table 1). *Neisseria* sp. was absent in antibiotic-treated CRS samples, but, due to the sample size, no significant association was confirmed (Fisher test *p* > 0.99; Appendix A).

Next, we investigated all samples from control carriers and CRS that contained *S. aureus*. The presence of *S. aureus* was significantly reduced in swabs simultaneously containing *Streptococcus* sp. (*p* = 0.003; Fisher test) and *Corynebacterium* sp. (*p* = 0.0009; Fisher test) and almost significantly with coagulase negative staphylococci (*p* = 0.055, Fisher test) (Table 2). These results suggest that these species interfere with *S. aureus* regardless of the origin of the sample. These interactions have already been described in other works summarized by Nair et al. [17].

In addition, 10 and 24 swabs, obtained from CRS and control donors, respectively, were analysed by shotgun metagenomic sequencing (Figure 1B).

We used MetaPhlan2 for taxonomic profiling, and although the relative abundance of the different species appeared to change between the two groups, no significant differences were found due to variations between samples (Figure 1B; Appendix A, indicator value, IndVal). Interestingly, at the genus level, *Staphylococcus* sp. dominated the disease sample (Figure 1B; 60% relative abundance, RA) compared to controls (40% RA). On the other hand, genera such as *Propionibacterium* sp. (22%) and *Corynebacterium* sp. (7%) were highly abundant in control samples but not in patients (Figure 1B). No differences were found between the two groups with respect to obligate anaerobic species. 

Subsequently, we measured alpha diversity indices such as Shannon, Simpson and evenness at the genus level and found no significant difference between control and disease samples (Wilcoxon rank sum test, Appendix A). We also measured beta diversity at the genus level (PERMANOVA, *p* = 0.2, Appendix A) based on the Bray–Curtis similarity. However, this test showed no significant differences between control and disease samples.

### 2.3. CRS and Control Microenvironments Were Distinguished by Metabolic Pathways

We then used HUMAnN2 for the functional profiling of the microbial communities. At the genus level, we observed no significant difference in alpha or beta diversity between control and diseased samples. Interestingly, at the functional level, 61 pathways were significantly different between control and CRS samples (Figure 2, heat map, IndVal). Pathways such as carbohydrate transport metabolism, ATP synthesis, cofactors and vitamins, photosynthesis and transcription were highly enriched in CRS. In contrast, pathways related to lipid metabolism were more representative in the control microbiome (Figure 2, heat map, IndVal).

### 2.4. S. aureus Populations of Patients and Controls Did Not Differ

*S. aureus* is the major microorganism colonizing the nasal sinuses and one of the primary pathogens in CRS. To characterise *S. aureus* isolates from control and CRS donors, a microarray was performed to detect specific genes and to assign isolates to clonal complexes (CCs) (Table 3, Table 4 and Table 5). One hundred *S. aureus* isolates (49 control and 51 CRS) were studied. The isolates belonged to a total of 14 different CCs. The distribution of CCs among control and CRS isolates was very diverse, and the main CCs were CC7, CC30 and CC45. Interestingly, CC15 was present in the CRS group. However, no significant differences were observed with regard to the origin of the sample (Table 5).

### 2.5. S. aureus Isolated from CRS Donors Induced Less Cytotoxicity and Glycolysis in Host Cells than Control Strains

Using functional metagenomics data, we investigated whether *S. aureus*, the primary CRS pathogen, responded differently to the nasal environment in CRS compared to controls. We tested whether staphylococcal isolates from both groups differed in virulence. Four isolates each from control (MiN) and CRS (CSS) patients were selected: CC7 (MIN 142 and CSS60), CC30 (MIN 53 and CSS74), CC45 (MIN 152 and CSS81) and CC15 (MIN 93 and CSS126) (Figure 3). The quantification of cell death showed higher cytotoxicity induced by control isolates compared to CRS strains within each CC, although not all differences were significant due to measurement variability (Figure 3A,B). In addition, the functionality of Agr, the major quorum-sensing regulator of staphylococcal virulence, was investigated using the CAMP test (Appendix A) [25,26]. IL-6 release, the main marker of the inflammatory response in *S. aureus* infection, was measured by ELISA after 24 h (Figure 4) [27]. A correlation was found between the induction of cell death and the expression of Agr. Control strains showed higher IL-6 and Agr induction than CRS strains (Figure 4 and Appendix A).

The activation of cell death has been linked to the induction of glycolysis [22]. Therefore, we assessed the metabolic response of A549 cells under different conditions using a Seahorse analyser. Glucose was added to stimulate metabolism, oligomycin suppressed oxidative phosphorylation and deoxyglucose (2-DG) inhibited glycolysis. Notably, differences in metabolic activity were observed between the strains, suggesting different effects on cell metabolism (Appendix A). Additionally, differences between control (MIN) and CRS strains were analysed for each pair. Significant differences in glycolysis induction were observed for all pairs, with control strains inducing higher glycolytic responses compared to CRS strains (Figure 5; Appendix A). Furthermore, no changes in ECAR were observed with bacteria alone (Appendix A), suggesting that the metabolic changes were mainly related to infection with A549.

Taken together, the cytotoxicity and glycolysis analyses suggested that CRS isolates were more adapted to host cells than control strains. This adaptation may be an important aspect in the pathogenesis of CRS.

### 2.6. Cyotoxicity Induced on Host Cells by Control Strains Changed with Serial Passages

We then investigated the effect of in vitro passages on pathogenicity in terms of cell death. We performed daily passages of *S. aureus* strains from CRS (CSS66, CSS74, CSS81 and CSS126) and control donors (MIN142, MIN53, MiN152 and MIN93) in fresh TSB media for 15 days (passages 1 to 15). Compared to the ancestral isolates, the cytotoxicity induced by the control strains decreased significantly after 15 passages. However, the respective cytotoxic properties were similar for both the ancestral and 15-passage isolates of the CRS strains (Figure 6).

## 3. Discussion

CRS is a multifactorial disease, which complicates its diagnosis [2]. Dysbiosis in the nasal microbiome has been associated with the development of CRS [28]. In our study, we compared nasal swabs from CRS patients and controls. The nasal microbiota was assessed by conventional culture methods on aerobic and anaerobic agar. As in previous studies, bacterial species showed no significant differences between the CRS and control groups [29]. However, *Neisseria* sp. was notably more abundant in the control group. Its susceptibility to antibiotics [24] raised questions about its reduced presence in CRS samples. Due to the limited antibiotic samples, no significant association was found between pre-/perioperative antibiotic treatment and the absence of *Neisseria* sp.

Regardless of the origin of the samples, we observed a negative correlation between the presence of *S. aureus* and other bacteria, such as *Streptococcus* sp., *Corynebacterium* sp. and coagulase-negative *Staphylococcus* sp. These results are consistent with previous studies [30], where a competitive interaction between these microorganisms was observed.

CRS and control samples were subjected to whole-genome shotgun metagenomics to uncover non-cultivable sinus bacteria [31]. While there were no significant taxonomic differences between the groups (CRS vs. control), the relative abundance of species appeared to change. In contrast, *Propionibacterium* sp. (22%) and *Corynebacterium* sp. (7%) were significantly present in control but not CRS samples, consistent with other nasal microbiome studies [10]. In addition, we did not find significant differences in the alpha and beta diversity indices between the two groups, which may have been due to the small cohort size or the sampling techniques. 

Although the diversity of the bacterial community appeared to be low, interestingly, functional profiling revealed significant differences between CRS and control samples. Sixty-one pathways showed notable differences. The carbohydrate transport metabolism, ATP synthesis, cofactors and vitamins, photosynthesis and transcription pathways were abundant in CRS. Despite the presence of photosynthetic pathways, *Cyanobacteria* sp. was not detected [32], suggesting that other non-cultivable species could produce these metabolites. Conversely, lipid metabolic pathways were more prominent in the control microbiome. This suggests that while the bacterial communities may be taxonomically similar, the nasal microenvironments differ in metabolites, which may contribute to greater inflammation in CRS compared to controls [5].

*S. aureus*, a prominent member of the nasal microbiota and a major pathogen associated with CRS, is strongly associated with invasive disease [16]. However, studies have not directly gauged the invasive *S. aureus* infection risk from CRS. Some suggest that CRS patients have higher nasal *S. aureus* colonization, which may increase the risk of invasive disease [33]. In addition, CRS patients often have a significant staphylococcal biofilm prevalence, which may increase the risk of invasive *S. aureus* infection [34,35]. However, the relationship between *S. aureus* colonization and invasive disease is complex and influenced by factors such as the host immune response, comorbidities and bacterial virulence [36,37,38].

According to the shotgun metagenomics data, the microenvironment of the nasal cavity was different in the control group and in CRS. Therefore, this environment could be responsible for *S. aureus* being colonized in one case and pathogenic in the other. The microbiome in control samples showed an increase in lipid metabolism pathways. *S. aureus* can use the long-chain fatty acids (AFAs) present on the skin and mucosal surfaces as a nutrient source and for outer membrane synthesis [39]. Furthermore, *S. aureus* is able to evade the antimicrobial properties of AFAs by sequestering host-specific AFAs in membrane vesicles (MVs). This allows *S. aureus* to colonise the nose [40,41]. In contrast, the microbiome in CRS patients was characterised by the activation of carbohydrate transport metabolism, the synthesis of ATP and cofactors and vitamins. Competition for glucose among all microorganisms is suggested by the activation of carbohydrate transport metabolism. Glucose plays an important role in the regulation of the Agr system, the main quorum-sensing regulator of virulence factors in *S. aureus*. Several studies have shown that the presence of excess glucose in hyperglycaemic abscesses increases the virulence potential of *S. aureus*, leading to poorer infection outcomes [42]. The uptake of glucose by *S. aureus* is influenced by CcpA, a regulator of central carbon metabolism. Assays with ΔccpA mutants have shown reduced Agr activity, suggesting that low glucose availability affects Agr system activity and subsequently virulence factor production in *S. aureus* through CcpA and ATP levels [42,43]. Therefore, CRS may cause *S. aureus* to down-regulate virulence factors, thereby promoting its persistence [44,45]. 

During the course of infection, dynamic metabolic shifts occur in both the host and staphylococci that influence the clearance process. *S. aureus* orchestrates these shifts by adjusting toxin expression via key regulatory genes such as *agr*, *sarA* and *sigB*, particularly during the transition from acute to chronic infection [21,44,46,47,48,49]. This transition transforms the host–*S. aureus* interaction into a form of nutrient competition that shapes the pathogen’s survival strategies within the host environment. *S. aureus* has the ability to induce different forms of cell death, including apoptosis, regulated necrosis (necroptosis) and pyroptosis [50]. While apoptosis exposes the pathogen to extracellular immune defences without causing inflammation, necroptosis and pyroptosis induce robust inflammatory responses that affect immune cell trafficking. In particular, pyroptosis has been shown to be effective in pathogen clearance, in contrast to necroptosis [21,50,51]. Our functional analysis revealed the pronounced induction of cytotoxicity in control isolates co-cultured with lung cells compared to CRS isolates. Given the link between cell death and host glycolysis [36,46,50], we used the Seahorse assay to assess the metabolic activity induced by *S. aureus* isolates on A549 cells. While all strains induced glycolysis, control-strain-infected cells showed significantly higher lactate production than CRS-infected cells. Host glycolysis is closely linked to pyroptosis and triggers a potent inflammatory response, together with the increased production of reactive oxygen species (ROS), which are critical for the immune response and cell death [21,50]. This mechanism predominates in the acute phase and facilitates the dissemination of *S. aureus* to other tissues [36], which may be similar to the behaviour of control strains. Conversely, CRS strains exhibit a distinct metabolic profile characterized by reduced glycolysis and increased oxidative phosphorylation. This reduction in glycolysis may deprive host cells of energy and prevent effective defence against *S. aureus*. In chronic infections associated with staphylococcal biofilms, *S. aureus* promotes an anti-inflammatory phenotype in macrophages, possibly through lactate transport into immune cells and the subsequent modulation of gene expression, thereby facilitating bacterial persistence [52]. This mechanism may occur in CRS and allow *S. aureus* to persist in the nose. In addition, *S. aureus* from CRS strains may induce necroptosis, a caspase-independent cell death mechanism that does not eradicate the pathogen [50]. However, further studies are warranted to elucidate the host modulation in control and CRS staphylococcal isolates.

These findings highlight the contrasting capabilities of control and CRS *S. aureus* strains: while control strains excel at killing host cells, CRS strains prioritise maintaining cell integrity and promoting bacterial persistence by modulating glycolysis and cell death pathways.

Subculture experiments were used to investigate the phenotypic flexibility of the isolates. Four isolates from each group were subcultured 15 times in fresh TSB. We then analysed the cytotoxicity of each isolate (ancestral and subcultured). Interestingly, ancestral isolates from control donors induced higher cytotoxicity than 15-passaged derivatives. However, the cytotoxicity of the ancestral isolate from CRS patients and its serially passaged forms showed no differences. This may have been due to the more challenging local environment in healthy controls, promoting reversible virulence up-regulation by immune cells and competing bacteria. If this were the cause, the pathogenesis of CRS would be determined by a host factor (or lack thereof) that allows the pathogen to cause relatively unchecked damage.

Infections position the nose as the starting point for *S. aureus*, leading to potentially life-threatening infections in multiple tissues [16]. The transition of *S. aureus* from a coloniser/commensal to a pathogen is controlled by host factors. Within the patient, mutation-driven evolution occurs, providing this pathogen with enhanced dissemination and colonisation capabilities for new infections [53]. This occurs mainly during acute infection. However, constant toxin expression requires energy and activates host immune responses to eliminate invaders [54]. To survive, *S. aureus* modifies virulence factors for host cell entry and evasion, a common scenario in chronic infections such as CRS [55].

Our study had limitations that necessitate the cautious interpretation of the data. The metagenomics quality control excluded many samples, leaving 34 out of 100 for analysis. This limitation may have confounded the detection of species/pathways or differences between CRS and controls. Furthermore, there may have been insufficient representation of the microbial diversity within the ecosystem being studied. As a result, the analysis may have failed to detect subtle taxonomic differences or associations that existed within the population, and it might have exaggerated individual or random properties of the samples considered. In addition, participants did not have known immunodeficiencies or use immunosuppressive medications. However, the limited assessment of the control and CRS groups (particularly in terms of allergies and allergy-related diseases, as shown in Table 1) may have missed or overlooked some immunological conditions and underestimated the inherent variability. In addition, all assays were performed using A549 cells. While A549 cells provide an established in vitro model [56,57,58,59], further work with sinonasal epithelial cells is needed to closely mimic the in vivo scenario. 

Although this was a single-centre study with a small metagenomic cohort, it highlighted that the differences between the CRS and control groups were not primarily related to differences in the bacterial microbiome of the nasal sinuses. Instead, they appeared to be rooted in microenvironmental shifts due to the differential metabolite expression by nasal sinus bacteria. These metabolic shifts may influence the immune response and the expression of virulence factors of *S. aureus* promoting host adaptation. In essence, CRS is a complex condition in which the microbiome plays a critical role, not necessarily in microbial dysbiosis but in alterations of the micronasal environment. Host-induced metabolic changes by colonisation and CRS strains drive different levels of glycolysis activation, enabling *S. aureus* to disseminate or persist intracellularly, thereby promoting the development of CRS.

## 4. Materials and Methods

### 4.1. Study Design and Patient Enrollment

This was a prospective observational study approved by the Ethics Committee of the University Hospital Jena (no. 2018-1175-Material). Participants gave informed consent before participation. Adult controls without nasal/paranasal disease, who had undergone unrelated head and neck surgery, were included. Adult CRS patients were recruited from those requiring endoscopic sinus surgery according to the German guidelines [60]. Briefly, surgery for CRS was indicated (1) in the case of a lack of symptom improvement following a sufficient drug therapy with a sufficiently probable correlate of the symptoms in endoscopy and/or imaging; or (2) if a conservative treatment attempt was not promising, (3) not possible or (4) not desired. CRSwNP patients were graded using the Malm score [61]. Treatment was not influenced by the study. Exclusion criteria were pregnancy, lactation, tumours, immunodeficiency, cystic fibrosis, uncontrolled diabetes, recent antibiotics and the acute bacterial exacerbation of CRS.

### 4.2. Sample Collection

Participants were orally intubated. Nasal swabs were taken before surgery, stored at −80 °C and analysed for bacteria at the Institute of Medical Microbiology. A total of 374 samples were processed, while 50 underwent shotgun metagenomic sequencing at BGI Tech Solutions. 

### 4.3. Metagenomic Sequencing:

Fifty samples from each group underwent shotgun metagenomic sequencing at BGI Tech Solutions (Hong Kong, China). Only 10 CRS patient swabs and 24 control swabs passed quality control for further analysis. Exclusion criteria included insufficient quantities, degradation, contamination affecting library construction and RNA interference.

Metagenomic sequencing was performed at BGI Tech Solutions (Hong Kong, China) as described in Qin et al. [62]. DNA from all swabs was extracted using a QIAamp PowerFecal Pro DNA Kit (Qiagen, Hilden, Germany). Then, library construction was performed at BGI using the KAPA kit (Roche, Shangai, China) following a series of steps: fragment genomic DNA, end reparation, the addition of A-tailing to the 3′ end, adapter ligation, fragment selection, PCR amplification and product purification and QC. Sequencing was performed on the Illumina XTen for PE150 (Illumnina, San Diego, CA, USA).

### 4.4. Metagenomics Analysis/Data Processing

*Quality control of sequence data*: Quality control to remove low-quality reads was performed as described previously [63]. Briefly, all Illumina primer/adapter/linker sequences were removed. Subsequently, low-quality regions (consecutive regions with Phred quality < 20) were trimmed. Finally, all reads were mapped to the human genome with BWA version 0.7.4 [64], and reads with >95% identity and 90% coverage were removed as human DNA contamination.

Taxonomic profiling: Taxonomic annotation of the high-quality reads was performed using MetaPhlAn2 version 2.7.7 [65] with default settings, generating taxonomic relative abundances. Bacterial community profiles were constructed at the phylum, genus and species levels for further analyses.

Functional annotation: The HUMAnN2 pipeline [66] version 0.11.2 was used for the functional annotation of the high-quality reads after quality control. The quantified pathway and gene family abundances in the unit of reads per kilobase (RPK) were then normalised to copies per million (CPM) units by the provided HUMAnN2 script, resulting in transcript-per-million-like (TPM) normalisation. Gene families were then regrouped to Pfam domains for further analyses.

### 4.5. Cells

All assays in this study were performed using A549. The alveolar epithelial cell line (A549; ATCC CCL-185) is a standardised, consistent cell line that can be cultured under controlled conditions. This cell line is well established in rhinosinusitis and *S. aureus* studies [56,57,58,59]. The advantages of using this cell line instead of primary nasal cells include the ease of culturing and maintenance and increased proliferative capacity, homogeneity and genetic stability compared to primary nasal cells.

### 4.6. Cell Death Induction

Four *S. aureus* isolates from control (MiN) and four from CRS (CSS) patients were selected from the main cluster complex (CC): CC7 (MIN 142 and CSS60), CC30 (MIN 53 and CSS74), CC45 (MIN 152 and CSS81) and CC15 (MIN 93 and CSS126).

The staphylococcal isolates were cultured in tryptic soy broth (TSB) at 37 °C, diluted and cultivated for 3 h. Bacteria were pelleted, washed and suspended in PBS (OD = 1 at 578 nm) and stored at −80 °C. Viable bacteria were confirmed by blood agar plating. A549 lung cells (ATCC: CCL-185) were seeded and infected with *S. aureus* (MOI, multiplicity of infection = 100) for 90 min; extracellular bacteria were killed with lysostaphin; and the cells were cultured for 24 h. After 24 h, the cells were detached and stained with 10 μg/mL PI, and flow cytometry was used for PI analysis (BD Accuri™ C6, Becton Dickinson GmbH, Heidelberg, Germany).

The MOI = 100 was chosen after preliminary experiments in our group where we tested different MOIs. Cells infected at MOI = 100 were able to internalise the maximum amount of bacteria with a weak cytotoxic effect. In addition, this MOI was suggested by other studies [67,68].

### 4.7. Measurement of IL-6 by ELISA

Cells were infected as described above, and, 24 h post-infection, the supernatants were collected and centrifuged at 1000 rpm for 5 min to exclude cellular derivatives. The ELISA for IL-6 (DuoSet ELISA, 5 Plates from R&D Systems, Bio-Techne GmbH, Wiesbaden-Nordenstadt, Germany) was performed following the manufacturer’s instructions. IL-6 secretion was measured by a microplate reader at 450 and 570 nm (TECAN Infinite^®^ 200 PRO).

### 4.8. CAMP Test: Assessment of Agr Activity on Sheep Blood Agar Plates

The CAMP test was used to assess the Agr activity of *S. aureus* by streaking test samples perpendicularly to an *S. aureus* strain that produced only β haemolysin and analysing the resulting patterns of haemolysis for the distinct pattern of Agr-controlled δ haemolysin [26]. The *S. aureus* strains in this study, along with RN4220, were cultivated in 10 mL of TSB for 17–20 h at 37 °C with agitation. For RN4220, an overnight culture (ONC) was streaked in the centre of a blood agar plate and then incubated for 5 h at 37 °C. For the strains, ONCs were streaked vertically to RN4220, and the plate was subsequently incubated for an additional 24 h at 37 °C. After a total of 24 h of incubation, the plates were placed in a 4 °C environment for 1–2 h. Following this cooling period, haemolysis was visually inspected.

### 4.9. Extracellular Flux Analysis

A549 cells (15,000 per well) were seeded in Seahorse XF96 microplates (Agilent Technologies, Waldbronn, Germany) in DMEM (10% FCS, 1% penicillin/streptomycin; SIGMA, Taufkirchen, Germany) and incubated overnight (37 °C, 5% CO_2_). The sensor cartridge was calibrated overnight (37 °C).

On the infection day, cells were washed with XF base medium (2 mM glutamine), infected with *S. aureus* strains (MOI 100) and incubated for 3 h (37 °C, no CO_2_). Gentamicin (200 µg/mL) eliminated extracellular bacteria, and cells were incubated for 60 min (37 °C, no CO_2_). A549 cells were washed twice with XF base medium (2 mM glutamine). The extracellular acidification rate (ECAR) was measured with an XF96e analyser using the Glycolysis Stress Test Kit (Agilent Technologies, Waldbronn, Germany). Measurement cycle: 3 min mixing, 3 min data acquisition (12 points).

### 4.10. Serial Bacterial Cultivation

Four *S. aureus* strains from control (MiN) and four from CRS (CSS) patients were selected that belonged to the most common CCs: CC7 (MIN 142 and CSS60), CC30 (MIN 53 and CSS74), CC45 (MIN 152 and CSS81) and CC15 (MIN 93 and CSS126). Isolates were grown on blood agar plates; 1 was colony picked, dissolved in 50 mL TSB and incubated for 17 h (160 rpm, 37 °C). Fresh medium was applied every 17 h (1:100 dilution), with 15 culture–transfer cycles for all strains.

### 4.11. Microarray-Based Characterisation of S. aureus Isolates

*S. aureus* isolates were genotyped via microarray (FZMB, Bad Langensalza, Germany). The array covered 333 targets from around 170 genes, detecting virulence, resistance, typing markers and strain/clonal complex assignment. Protocols and sequences were previously described [69,70]. *S. aureus* was cultivated on blood agar, DNA was prepared and linear amplification with specific primers was performed, followed by biotin-16-dUTP incorporation. Hybridisation with array probes was then performed, followed by streptavidin horseradish peroxidase incubation and dye addition, causing local precipitation. Microarrays were photographed and analysed using an InterVision reading device (InterVision, Bad Langensalza, Germany). The selection of the *S. aureus* isolates was performed at random, with 49 control strains and 51 CRS strains being selected.

### 4.12. Statistical Analysis

Statistical analysis utilised IBM SPSS Statistics 19, R version 4.0.2 (SPSS Inc, Chicago, USA) and GraphPad Prism version 10.0.0 for Windows (GraphPad Software, Boston, MA USA, www.graphpad.com accessed on 9 February 2024) Demographics and baseline differences were assessed with Pearson’s chi-square and Mann–Whitney U-tests for age. Bacterial presence, co-isolation, clonal complexes and gene frequency were analysed with Fisher’s exact test. 

Alpha diversity (Shannon, Simpson) was calculated with the R package vegan [71]; differences were tested with the Wilcoxon rank sum test. To estimate community dissimilarities, Bray–Curtis distances were calculated using the R package vegan based on the relative genus abundance. Abundance comparisons used the indicator value (IndVal) labdsv package [72] for genera, pathways and Pfams with ≥10% prevalence. Additional tests were performed, including unpaired *t* tests, two-way ANOVA and Tukey’s post hoc test. 

## 5. Conclusions

To summarise, this pioneering study, although conducted in a single centre with a limited metagenomic cohort, sheds important light on chronic rhinosinusitis (CRS). Contrary to the conventional understanding, the differences between the CRS and control groups were not primarily due to variations in the nasal sinus bacterial microbiome. Instead, the study revealed a compelling sequence of microenvironmental changes driven by the distinct metabolite expression of nasal sinus bacteria.

These metabolic nuances, in turn, exert a significant influence on the immune response and the expression of virulence factors in *S. aureus*, fostering an environment conducive to host adaptation. In essence, CRS emerges as a complex interplay in which the microbiome plays a critical role, not in microbial dysbiosis per se but in orchestrating subtle changes in the micronasal environment.

The host-induced metabolic changes, orchestrated by both colonising and CRS strains, set the stage for different levels of glycolysis activation. This activation, intriguingly, enables *S. aureus* to either disseminate or persist intracellularly, ultimately contributing to the complex development of CRS. This study invites us to reconsider our understanding of CRS as a dynamic and multifaceted condition in which the microbiome, rather than causing dysregulation, plays a pivotal role in shaping the micronasal landscape.

## Figures and Tables

**Figure 1 ijms-25-02229-f001:**
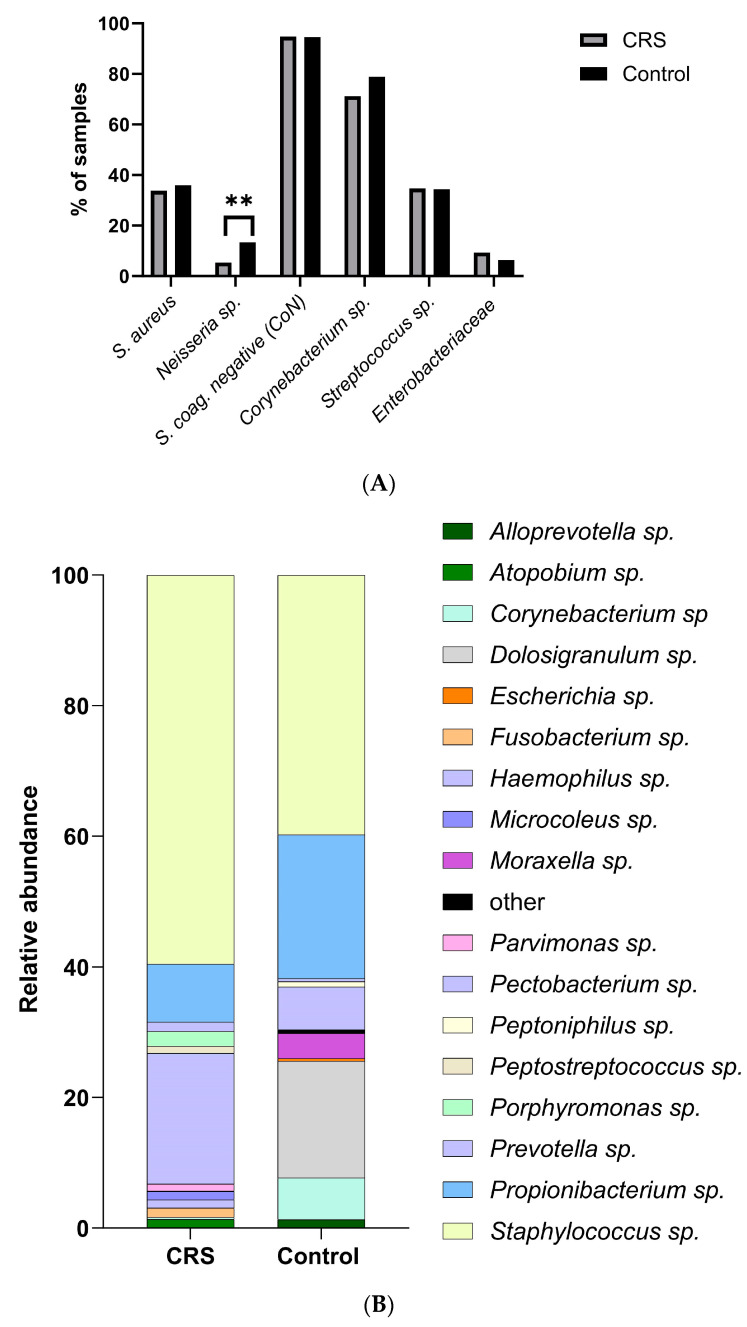
Distribution of the bacterial community in control and CRS samples. (**A**) Bacterial community composition at the species level of the nasal cavities of control and CRS patients was analysed by conventional culture-dependent techniques. Differences in the proportions between both groups were compared using Fisher’s exact test (** *p* < 0.01). (**B**) Sinonasal microbial composition of control and CRS samples was analysed by metagenomics. The composition of the bacterial community at the genus level is shown.

**Figure 2 ijms-25-02229-f002:**
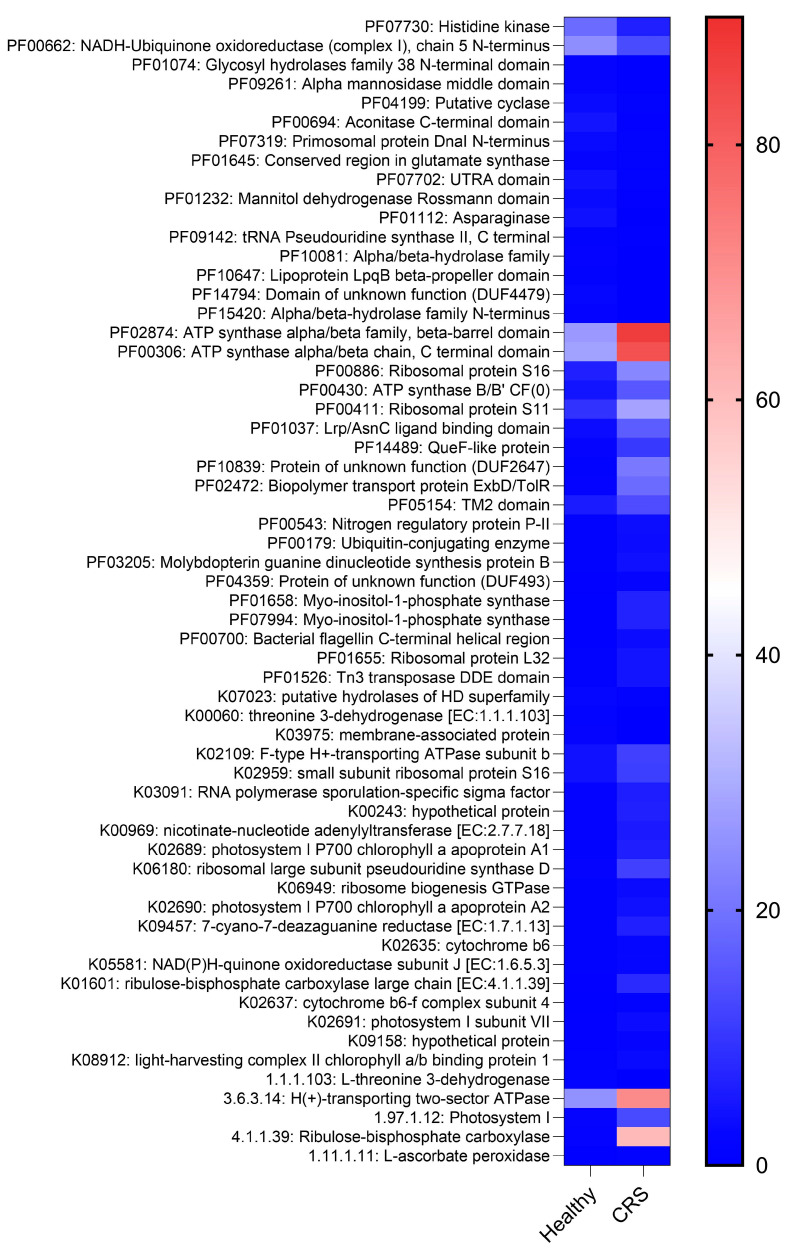
Metabolic profiles from control and CRS samples. The heatmap depicts metagenomic bacterial community functional annotation based on PFAM, KEGG orthologs and enzyme commission (EC) abundance for both groups. The heatmap’s colour gradient is based on the pathway enrichment in each group (high enrichment in red to low enrichment in blue) and the unit is transcripts per million (TPM).

**Figure 3 ijms-25-02229-f003:**
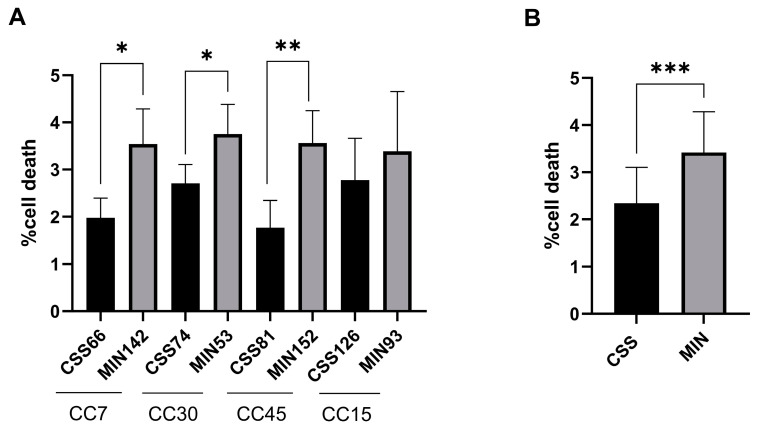
Cell death induction by *S. aureus* isolates from control (MIN) and CRS (CSS) donors. (**A**) Staphylococcal isolates from control and CRS donors from the main clonal complexes (CC7, CC30, CC45 and CC15) were selected. Human epithelial lung A549 cells (ATCC: CCL-185) were infected, and cell death induction was measured by staining the cells with propidium iodide after 24 h. (**B**) Proportions of dead cells obtained for staphylococcal isolates from the control and CRS groups. Differences between the proportions of dead cells infected with isolates from both groups were analysed by unpaired *t* tests (* *p* < 0.05, ** *p* < 0.01 and *** *p* < 0.001). Data are presented as the average of triplicate determinations, and error bars represent the standard deviation.

**Figure 4 ijms-25-02229-f004:**
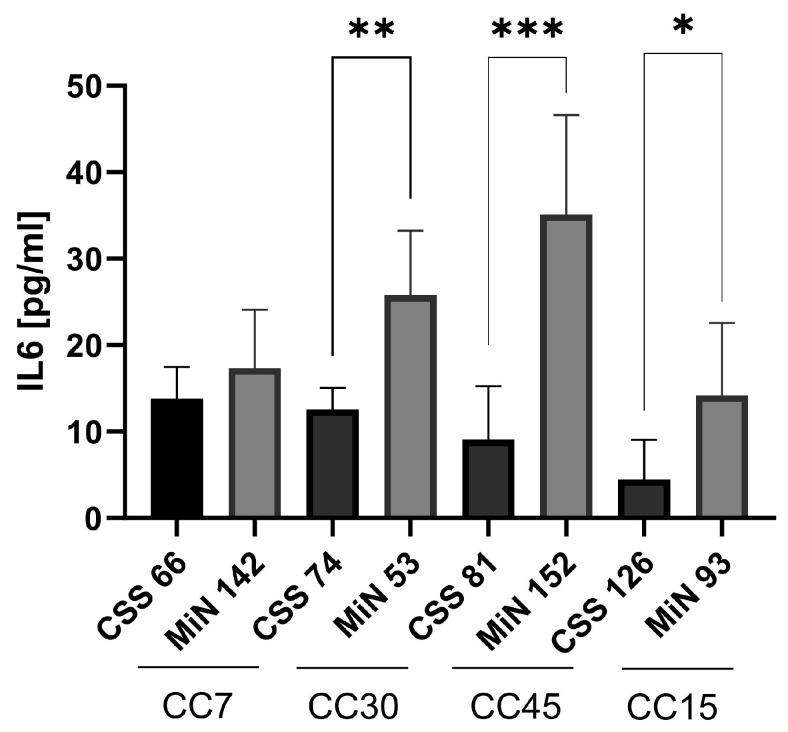
IL-6 induced by *S. aureus*-infected cells. Staphylococcal isolates from control and CRS donors were selected from the major clonal complexes (CC7, CC30, CC45 and CC15). Human lung epithelial A549 cells (ATCC: CCL-185) were infected and the supernatant was analysed by ELISA at 24 h post-infection. Differences in IL-6 release between the isolates from each pair infected were analysed by unpaired *t* tests (* *p* < 0.05, ** *p* < 0.01, *** *p* < 0.01). Data are presented as the mean of triplicate determinations, and error bars represent the standard deviation.

**Figure 5 ijms-25-02229-f005:**
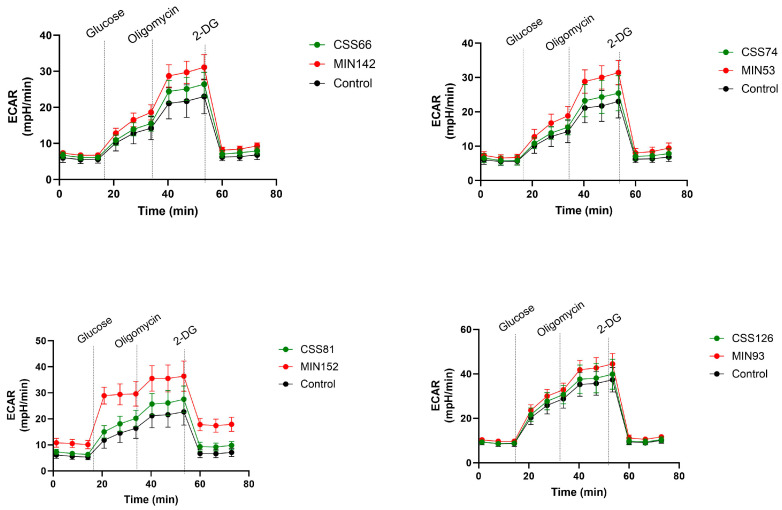
*S. aureus* isolates from the control and CRS groups induced glycolysis in different manners in A549 cells. Metabolic activity of A549 cells was monitored by measuring the extracellular acidification rate (ECAR) of uninfected and infected cells by Seahorse. A549 cells were infected with four different control and CRS *S. aureus* strains. The additions of glucose, oligomycin and 2-DG (deoxyglucosoe) are indicated. Statistical significance in the assay was compared to PBS alone and between the strains by two-way ANOVA (Appendix A).

**Figure 6 ijms-25-02229-f006:**
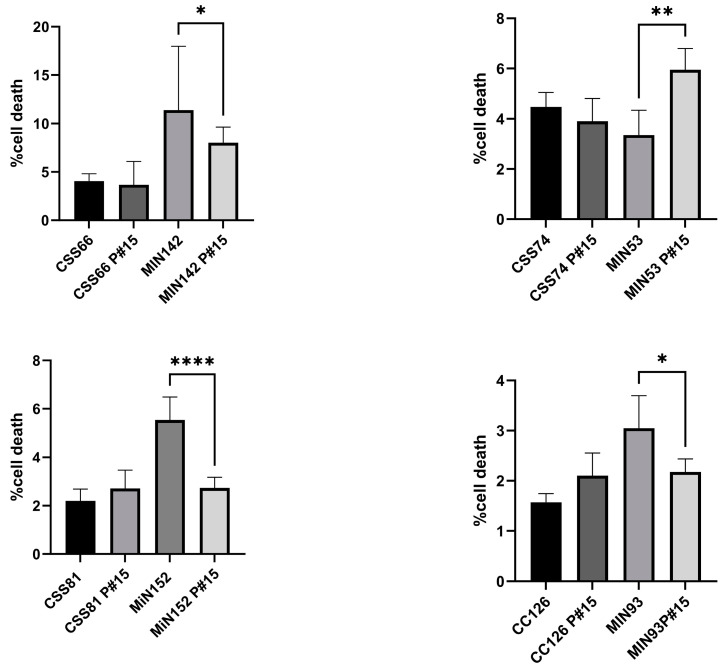
*Differential cytotoxicity induced by S. aureus isolates and their serially passaged strains*. Cytotoxicity induction of *S. aureus* isolates and serially passaged strains from staphylococcal isolates from the control and CRS groups on human epithelial lung A549 cells. Data are presented as the average of triplicate determinations, and error bars represent the standard deviation. Differences in cell death between the original and serially passed isolates from each pair were analysed by unpaired *t* tests (* *p* < 0.05, ** *p* < 0.01, **** *p* < 0.01).

**Table 1 ijms-25-02229-t001:** Volunteers/probands’ and patients’ characteristics.

	Control Volunteers (N = 128)		Patients with CRS (N = 246)		
Parameter	N	%	N	%	*p* *
Gender					0.595
Female	52	40.6	93	37.8	
Male	76	59.4	153	62.2	
Smoking					0.029
Yes	39	30.5	37	15.0	
No	89	69.5	209	85.0	
Allergy					0.812
Yes	53	41.4	107	43.5	
No	75	58.6	139	56.5	
N-ERD					0.002
Yes	0	0	18	7.3	
No	128	100	228	92.7	
Asthma					0.003
Yes	7	5.4	46	18.7	
No	121	94.6	200	81.3	
Diabetes					0.074
Yes	16	12.5	9	3.7	
No	112	87.5	237	96.3	
CRS type					NA
CRSsNP	NA	NA	166	67.5	
CRSwNP	NA	NA	80	32.5	
Pre- or perioperative antibiotic treatment					0.358
Yes	23	18.0	59	24.0	
No	105	82.0	187	76.0	
	Mean ± SD	Median, Range	Mean ± SD	Median, Range	*p* **
Age, years	53.1 ± 17.2	55, 18–89	49.7 ± 16.2	50, 18–83	0.108
MALM score	NA	NA	1.4 ± 1.1	2, 0–3	NA

* Pearson’s chi-square test; ** Mann–Whitney U-test; N-ERD = NSAID-exacerbated respiratory disease; NA = not applicable.

**Table 2 ijms-25-02229-t002:** Fisher test of co-isolation of *S. aureus* with other bacterial species (** *p* < 0.01; *** *p* < 0.001).

Bacterial Species	Fisher Test
*Corynebacterium* sp.	*p =* 0.003 **
*Staphylococcus coag. negative*	*p =* 0.055
*Streptococcus* sp.	*p =* 0.0009 ***
*Enterobacteriaceae*	*p* > *0.99*
*Neisseria* sp.	*p =* 0.42

**Table 3 ijms-25-02229-t003:** Virulence-associated markers of *S. aureus* from CRS cases and controls.

	CRS Patient Isolates, Number	CRS Isolates, Percent	Control Isolates, Number	Control Isolates, Percent
Species markers (*gapA, katA, coA, nuc, spa, sbi*)	51	100.0	49	100.0
*agr*-group I	33	64.7	32	65.3
*agr*-group II	8	15.7	6	12.2
*agr*-group III	9	17.6	11	22.4
*agr*-group IV	1	2.0	1	2.0
*hld*	51	100.0	49	100.0
*tst1*	7	13.7	12	24.5
*sea*	4	7.8	2	4.1
*sea (N315) = sep*	8	15.7	7	14.3
*seb*	1	2.0	1	2.0
*sec, sel*	7	13.7	4	8.2
*sed*	1	2.0	0	0.0
*see*	0	0.0	0	0.0
*seh*	2	3.9	5	10.2
*sej*	1	2.0	0	0.0
*sek, seq*	0	0.0	0	0.0
*ser*	1	2.0	0	0.0
*egc* genes *(seg, sei, selm, seln, selo, selu)*	27	52.9	27	55.1
*ORF CM14*	0	0.0	1	2.0
*lukF/S-hlg*	51	100.0	49	100.0
*lukF/S-PV*	0	0.0	0	0.0
*lukF-PV (P83)/lukM*	1	2.0	0	0.0
*lukD*	23	45.1	23	46.9
*lukE*	24	47.1	22	44.9
*lukX/Y*	51	100.0	49	100.0
*hla*	50	98.0	49	100.0
*sak*	39	76.5	35	71.4
*chp*	32	62.7	28	57.1
*scn*	49	96.1	43	87.8
*etA*	1	2.0	0	0.0
*etB*	0	0.0	0	0.0
*etD*	0	0.0	3	6.1
*edinA, edinC*	0	0.0	0	0.0
*edinB*	0	0.0	3	6.1
ACME	0	0.0	0	0.0
Capsule type 5	11	21.6	18	36.7
Capsule type 8	40	78.4	31	63.3
*cna*	29	56.9	22	44.9
*sasG*	15	29.4	15	30.6

**Table 4 ijms-25-02229-t004:** Resistance-associated markers of *S. aureus* from CRS cases and controls.

	CRS Patient Isolates, Number	CRS Isolates, Percent	Control Isolates, Number	Control Isolates, Percent
*mecA **	0	0.0	1	2.0
SCC*mec* Iva *	0	0.0	1	2.0
*mecC*	0	0.0	0	0.0
*blaZ*	29	56.9	37	75.5
*erm*(A)	3	5.9	3	6.1
*erm*(B), *erm*(C), *lnu*(A), *msrA*	0	0.0	0	0.0
*aacA-aphD*	0	0.0	1	2.0
*aadD, aphA3, sat*	0	0.0	0	0.0
*fusC, far1*	0	0.0	0	0.0
tet(K)	0	0.0	3	6.1
dfrA, mupR, tet(M), cat, cfr, fexA, van genes	0	0.0	0	0.0
*qacC*	1	2.0	0	0.0

* in a CC30-MRSA-IV (PVL-/tst1-).

**Table 5 ijms-25-02229-t005:** Clonal complex affiliations of *S. aureus* from CSS cases and controls.

	CRS Patient Isolates. Number	CRS Isolates. Percent	Control Isolates. Number	Control Isolates. Percent
CC5	1	2.0	1	2.0
CC7	7	13.7	7	14.3
CC8	3	5.9	4	8.2
CC15	7	13.7	4	8.2
CC20	1	2.0	0	0.0
CC22	2	3.9	2	4.1
CC25	0	0.0	3	6.1
CC30	8 *	15.7	7	14.3
CC34	1	2.0	4	8.2
CC45	12	23.5	8	16.3
CC50	1	2.0	0	0.0
CC97	0	0.0	3	6.1
CC101	2	3.9	0	0.0
CC182	1	2.0	1	2.0
CC188	2	3.9	1	2.0
CC398	3	5.9	3	6.1
ST848	0	0.0	1	2.0
TOTAL	51	100.0	49	100.0

* including one CC30-MRSA-IVa (PVL-/tst1-).

## Data Availability

The original contributions presented in the study are included in the article and Appendix A, further inquiries can be directed to the corresponding author.

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
