# Peer review of "Reduced Glycolysis and Cytotoxicity in Staphylococcus aureus Isolates from Chronic Rhinosinusitis as Strategies for Host Adaptation"

_ijms, 2024, doi:10.3390/ijms25042229_

Round 1

Reviewer 1 Report

Comments and Suggestions for Authors

Summary:

This manuscript investigates the microbiome and host-pathogen interactions in chronic rhinosinusitis (CRS). The authors analyzed nasal swabs from CRS patients and controls using culture, metagenomics, and functional assays with Staphylococcus aureus isolates.

Key findings:

1.      No major differences in microbiome composition between CRS and controls based on culture and metagenomics, except reduced Neisseria in CRS likely due to antibiotics.

2.      2. Metagenomics showed differences in metabolic pathways between groups, with more carbohydrate metabolism in CRS and more lipid metabolism in controls.3.

3.      S. aureus isolates showed no major genetic differences between CRS and controls based on microarray.

4.      However, functional assays found CRS isolates were less cytotoxic and induced less glycolysis in lung cells compared to control isolates.

5.      Control isolates also lost cytotoxicity after repeated passage, whereas CRS isolates remained stable.

This study provides valuable insights into the role of the microbiome and host-pathogen interactions in CRS. A key strength is the combined use of culture, metagenomics, and functional assays to assess not just compositional differences but also functional capabilities between CRS and control microbiota. The findings challenge the notion that microbial dysbiosis drives CRS, instead pointing to complex microenvironmental factors influencing S. aureus adaptation and virulence. Limitations include the small metagenomics cohort and lack of participant clinical details. Overall, this is a pioneering study highlighting the intricate interplay between host and microbiome in CRS pathogenesis.

Suggestions:

1.      Expand discussion of metagenomics results. Are there limitations due to the small sample size that may have prevented detection of taxonomic differences?

2.      Discuss relevance of metabolic pathway differences found. Do these suggest microenvironmental changes that could influence virulence?

3.      Provide more context for the differences in cytotoxicity and glycolysis induction between CRS and control S. aureus isolates. What is the significance?

4.      Provide more details on participant characteristics in Table 1 (e.g. age ranges, gender split per group).

Reviewer 2 Report

Comments and Suggestions for Authors

Dear Authors

Thanks for your manuscript with title: Reduced Glycolysis and Cytotoxicity in Staphylococcus aureus Isolates from Chronic Rhinosinusitis as strategies for host adaptation. The review of the aforementioned manuscript has been finished and it sounds interesting, however there are some points about it that must be addressed.

Best Regards

Reviewer 3 Report

Comments and Suggestions for Authors

This study is very unique and challenging work, utilizing with so many molecular and metabolic biology techniques, to seek the contribution of S. aureus for the pathogenesis of chronic rhinosinusitis. I would like to highly evaluate this work and recommend to be accepted for publication.  Actually speaking, as authors self recognize in discussion, their research data could be argued from many individual aspects. However, their striking data indicated  that reduced glycolysis and cytotoxicity of infected respiratory epithelial cells induced infected S. aureus obtained from CRS  patients might be attributed to the persistence of chronic inflammation in sinonasal epithelial cells. Therefore, I do recommend this manuscript can be accepted and cited to this journal as soon as possible.  A couple of question is the followings; 

1. A549 cells are not sinonasal mucosal epithelial cells, did not use sinonasal epithelial cells for in vitro S. aureus infection model to see glycolysis and cytotoxicity assay ?  

2. S. aureus role in CRS has been discussed in many previous articles by Claus Bachert, Mitsuhiro Okano, and others. You should cite those papers in reference, don't you.

3. Diagnosis of Chronic rhinosinusitis was made sure according to the German guideline, but more precise description should be cited in methodology. 

Reviewer 4 Report

Comments and Suggestions for Authors

I find the study very interesting, but it could be improved, I suggest the following changes.

1. What are those factors that influence the virulence of S. aureus that promote its adaptation to the nasal environment during CRS and that were not included in the abstract?

2. In the introduction they cannot only refer to the fact that CRC affects 10% of the population of Europe, the authors must talk about the world population and then they can be more specific with the population of Europe.

3. In the introduction I recommend placing the name of the microorganism followed by spp. and not sp. (This is because spp. refers to several species within the same genus or the species it refers to is not known).

4. In lines 56-57, how relevant is S. aureus in patients with CRS, since they give % only for healthy patients.

5. The authors are required to mention the limitations and perspectives of the study.

6. I think the word summary in the conclusion should be changed, since the summary is at the beginning of the paper.

7. The order in which the authors present the paper is not correct. The article should be structured as follows: 1. Introduction, 2. Materials and methods, 3. Results, 4. Discussion, 5. Conclusions, 6. Limitations and perspectives of the study.

8. In the first paragraph of the results, the authors should mention both the % of men and women, and not just state that there was more male participation (62%).

9. When the authors say that smokers were more frequent in the control group, it would also be worth mentioning whether they were more men or women and the %.

10. The authors could improve the results by mentioning, in addition to the differences between the two groups (control and CRC), the differences that could exist between genders.

11. Table No. 1 should improve, taking into account the gender of the patients.

12. The names of the bacteria in the graphs must be corrected.

13. Write in italics the names of the genus and species of the bacteria.

Round 2

Reviewer 2 Report

Comments and Suggestions for Authors

Dear Authors

Thanks for your revised manuscript.

I checked the manuscript and almost, all the comments has been addressed.

Best Regards

Reviewer 4 Report

Comments and Suggestions for Authors

The changes suggested to the authors were made, so it is considered that the article can be published